# Fast and effective molecular property prediction with transferability map
Shaolun Yao[1,2,3], Jie Song[3,4], Lingxiang Jia[2], Lechao Cheng[5], Zipeng Zhong[2], Mingli Song[2,3] & Zunlei Feng [3,4] ✉

Effective transfer learning for molecular property prediction has shown considerable strength in addressing insufficient labeled molecules. Many existing methods either disregard the quantitative relationship between source and target properties, risking negative transfer, or require intensive training on target tasks. To quantify transferability concerning task-relatedness, we propose Principal Gradient-based Measurement (PGM) for transferring molecular property prediction ability. First, we design an optimization-free scheme to calculate a principal gradient for approximating the direction of model optimization on a molecular property prediction dataset. We have analyzed the close connection between the principal gradient and model optimization through mathematical proof. PGM measures the transferability as the distance between the principal gradient obtained from the source dataset and that derived from the target dataset. Then, we perform PGM on various molecular property prediction datasets to build a quantitative transferability map for source dataset selection. Finally, we evaluate PGM on multiple combinations of transfer learning tasks across 12 benchmark molecular property prediction datasets and demonstrate that it can serve as fast and effective guidance to improve the performance of a target task. This work contributes to more efficient discovery of drugs, materials, and catalysts by offering a task-relatedness quantification prior to transfer learning and understanding the relationship between chemical properties.

Molecular property prediction, which involves identifying molecules with desired properties[1,2], poses a critical challenge prevalent across various scientific fields. It holds particular significance in chemistry for designing drugs, catalysts, and materials. In recent years, artificial intelligence (AI) technologies have come mainstream in this area, and AI-guided chemical design can efficiently explore chemical space while improving performance based on experimental feedback, showing promise from laboratory research to real-world industry applications[3]. However, it is common that the experimental data size is small as producing labeled data requires time-consuming and expensive experiments[4,5]. In contrast, transfer learning[6] has become a powerful paradigm for addressing data scarcity problem by exploiting the knowledge from related datasets across fields such as natural language processing[7,8], computer vision[9,10], and biomedcine[11,12]. In chemistry, transfer learning leverages pre-trained models on extensive or related datasets to facilitate efficient exploration of vast chemical space[13,14] for

various downstream tasks. It has been used to predict properties[15,16], plan synthesis[17,18], and explore the space of chemical reactions[19–22].

Transfer learning can enhance molecular property prediction in limited data sets by borrowing knowledge from sufficient source data sets, thus improving both model accuracy and computation efficiency. Although several previous works have explored the power of transfer learning to enhance molecular property prediction[11,12,23–25], challenges remain. One major challenge is negative transfer, which occurs when the performance after transfer learning is adversely affected due to minimal similarity between the source and target tasks[26,27]. For example, Hu et al.[23] observed that pretrained GNN (at both node-level and graph-level) performed well but yielded negative transfer when pretrained at the level of either entire graphs or individual nodes. Additionally, some supervised pre-training tasks unrelated to the downstream task of interest can even degrade the downstream performance[23,28].

[1]Collaborative Innovation Center of Artificial Intelligence by MOE and Zhejiang Provincial Government, Zhejiang University, 310027 Hangzhou, China. [2]College of Computer Science and Technology, Zhejiang University, 310027 Hangzhou, China. [3]Shanghai Institute for Advanced Study of Zhejiang University, 201203 Shanghai, China. [4]School of Software Technology, Zhejiang University, 315048 Ningbo, China. [5]School of Computer Science and Information Engineering, Hefei University of Technology, 230009 Hefei, China. ✉e-mail: zunleifeng@zju.edu.cn

Negative transfer primarily stems from suboptimal model and layer choices, as well as insufficient task relatedness, highlighting the need to evaluate transferability prior to applying transfer learning. In computer vision, some researchers have recently focused on selecting the best model from a pool of options by estimating the transferability of each model[29–32]. In molecular property prediction, recent efforts involve investigating the relatedness of the source task to the target task. To maximize the performance on a target task and prevent negative transfer, existing methods mainly rely on a molecular distance metric to measure the similarity of molecules, such as Tanimoto coefficient (based on molecular fingerprint)[33,34] and a chemical distance measure (based on fingerprint and subgraph)[35]. Moreover, inspired by the seminal work of Taskonomy[36] and subsequent works such as RSA[37], an emerging approach in modeling the similarity between computer vision tasks, there have been some recent attempts towards developing representation similarity measurement to quantify similarity between biological compounds, such as molecules[38,39], proteins[40], and macromolecules[41]. These techniques leverage pre-trained model representations to assess biological compound similarity across different tasks. As they stand in need of model optimization across all datasets, and their computational costs that are as high as fine-tuning with target tasks exclude their applications in quantifying transferability prior to fine-tuning. In addition, there is still no metric for quantifying how suitable the source property is for the target property prior to training on the target task in a computation-efficient manner.

To this end, we put forward a simple, fast, and effective Principal Gradient-based Measurement (PGM) to quantify the transferability from the source property to the target property (Fig. 1). First, to approximate the direction of model optimization on a molecular property prediction dataset, we design a restart scheme to calculate a principal gradient in an optimization-free manner. The distance between the principal gradient obtained from model training on the source dataset and that derived from the target dataset indicates transferability. Second, we build a quantitative transferability map by performing PGM on various molecular property prediction datasets to show the inter-property correlations in property space distribution. Third, through the map, we can capture and transfer the most desirable source dataset for the given target dataset, so as to promote performance on the target task and avoid negative transfer. To verify the effectiveness of the proposed PGM, we evaluated PGM thoroughly on 12 benchmark datasets from MoleculeNet[42] with various molecular property prediction tasks. Comprehensive experiments on multiple combinations of transfer learning tasks demonstrate that the quantitative transferability derived from PGM is strongly related to the transfer learning performance. The proposed approach can serve as fast and effective guidance to enhance

the transfer performance of molecular property prediction. Our contributions can be summarized as follows:

- We propose a molecular property transferability measurement technique, termed as PGM, which can rapidly and effectively measure the transferability between the source and target molecular property prediction datatsets.
- Furthermore, we perform PGM on molecular benchmarks to build a transferability map that quantifies inter-property correlations. The map is extensible and can be a reference standard for transfer learning in molecular property prediction, even when applied to a few target samples.
- We empirically demonstrate that the transferability measured by the proposed technique has a strong correlation with the transfer learning performance of molecular property prediction tasks in reality.

Significantly, the proposed PGM has another two advantages: (1) it is computation-efficient, free of either calculating by brute force or model optimization. (2) it includes model-agnostic molecular encoder and predictor, thereby applied to various machine learning model frameworks.

## Results

### Overview of PGM guided transfer learning

In this paper, we introduce Principal Gradient-based Measurement (PGM) to quantify transferability between the source and target molecular properties. The framework of PGM guided transfer learning comprises three components as illustrated in Fig. 1: (1) PGM, (2) transferability map building, and (3) application of transfer learning.

**PGM.** The model optimization process aims to move from the initial point towards the optimal convergence point (Fig. 1a), where the model's parameters minimize the loss function and the model achieves peak performance. This optimization process involves iteratively updating the model parameters to reduce the loss function, commonly employing gradient descent as the preferred method. Building upon this conceptualization, the model is initialized from the same initial point and subsequently optimized on both source and target molecular property prediction datasets. The distance between the optimal convergence point of the source task and that of the target task refers to transferability. A smaller distance (between source property A and the target property, colored in orange and red, respectively) suggests a higher level of task similarity. Based on the invisible optimal convergence point, computing the transferability by optimizing models is as prohibitively expensive as training models on the given target dataset and all alternative source datasets, while the transferability offers benefits only when it can be

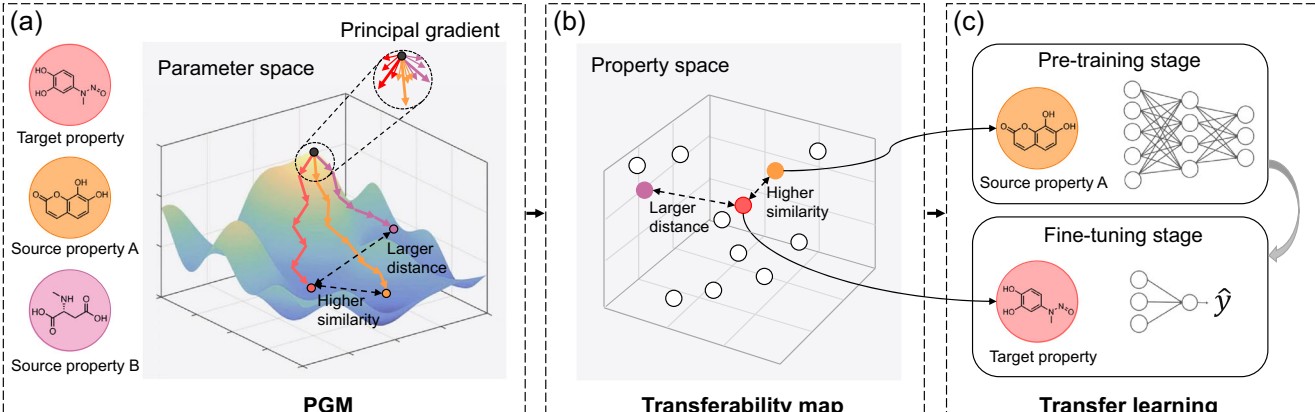

**Fig. 1 | Illustration of Principal Gradient-based Measurement (PGM) for guiding transfer learning in molecular property prediction. a** Firstly, we design an optimization-free scheme to calculate a principal gradient for approximating the direction of model optimization on a molecular property prediction dataset. PGM measures the transferability as the distance between the principal gradient obtained

from the source dataset and that derived from the target dataset. **b** Then, we perform PGM on various molecular property prediction datasets to build a quantitative transferability map indicating inter-property correlations. **c** Finally, through the map, we can effectively choose a source dataset with higher similarity to the given target dataset in transfer learning.

calculated as a prior. Thus, we are motivated to pursue a computation-efficient transferability measurement.

Gradients can capture intrinsic task-related characteristics, playing a strong predictive role in model optimization on a dataset, as evidenced by recent studies[43–46]. Grad2Task[43] posits that gradients can be used as features to capture task nature and distinguish between tasks under a meta-learning framework. Numerous gradient surgery techniques[44,45] have been proposed to address gradient conflict in multitask learning. In light of gradients being explored in the context of task-relatedness, we derive PGM, a simple measure based on gradients, to quantify the transferability concerning task-relatedness. Specifically, the proposed principal gradient, obtained through model re-initialization and gradient expectation calculation, approximately indicates the direction for model optimization on a dataset. Afterwards, PGM measures the transferability as the distance between the principal gradient obtained from model training on the source dataset and that derived from the target dataset.

**Transferability map building**. To visualize the task-relatedness between the source and target molecular property prediction datasets, we create a quantitative transferability map based on the distances between the principal gradients obtained from these datasets, shown in Fig. 1b. When assessing the transferability from potential source datasets to a given target dataset, the distance between source and target datasets serves as an indicator of their knowledge transfer ability. We utilize the PGM distance as the distance metric, which is measured by the difference between the principal gradients of model training on each pair of the datasets.

**Application of transfer learning**. To validate the effectiveness of the transferability map, we further explore transfer learning experiments in Fig. 1c. Given a target dataset, we select the most similar source dataset for pre-training. During the fine-tuning phase, we fix the feature extractor initialized from the pre-trained model, while train the predictor from scratch on the target task.

The proposed methodology is tested on multiple molecular benchmarks, and the results are described in the following sections.

## Transferability map built by PGM

Molecular properties of interest can vary widely in scale, ranging from macroscopic influences on the human body to microscopic electronic properties, such as toxicity to humans[47], the ability of drugs to permeate the brain[48], and hydration free energy[49]. We evaluate the task-relatedness of 12 benchmark datasets from MoleculeNet[42] in three categories: biophysics, physiology, and physical chemistry.

We build a quantitative transferability map to intuitively observe the task-relatedness between these molecular property prediction datasets, shown in Fig. 2. We perform PGM on each dataset to obtain its principal gradient. The pairwise PGM distances between the obtained principal gradients are used to compute the $12 \times 12$ transferability map, showing the transferability of these datasets in property space distributions.

## Transferability map-guided transfer learning improves performance

To investigate the effectiveness of the transferability map in measuring task-relatedness, we design a transferability map-guided cross-task transfer learning strategy. Specifically, each of the 12 datasets is used as the target dataset, while the remaining 11 datasets are employed as source datasets, as described below. Initially, the model is trained on each source dataset to obtain pre-trained models. Subsequently, each of these pre-trained models is fine-tuned on the target dataset, with learning curves shown in Fig. S1 of Supplementary Results. We refer to this experiment as the main experiment below. Due to page limit, the wall-clock time comparison between PGM and fine-tuning is available in Table S2 of Supplementary Results.

As depicted in Fig. 3, a significant correlation between the predicted transferability and the transfer learning performance across various tasks can be observed. As anticipated, when arranging the PGM distance in ascending order, the corresponding transfer performance overall exhibits a gradual decrease for the initial 9 classification tasks (ROC-AUC, higher is better) and a gradual increase for the final 3 regression tasks (RMSE, lower is better). Meanwhile, as the PGM distance increases, there is a decline in transfer performance observed in all 11 scenarios for BACE[50] and more than 8 scenarios for the other classification datasets. For the Tox21 source dataset, the knowledge transfer from ToxCast achieves more improvement in ROC-AUC than any other source dataset. Similarly, for the ToxCast source dataset, Tox21 outperforms 8 out of 10 source datasets in terms of improvement. This is expected because both Tox21 and ToxCast focus on exploring the toxicity of compounds to humans. Another interesting finding is that, among all source datasets, PCBA is a top-three performer in boosting the performance of almost all target datasets in classification scenarios, due to its diverse structure or useful out-of-distribution information. Additionally, despite belonging to distinct domains, some datasets exhibit higher task similarity, such as HIV, MUV, PCBA, Tox21, ToxCast, and SIDER. Collectively, these findings corroborate the utility of the transferability map and suggest that it can effectively assist in selecting the most similar source dataset for the given target dataset, thereby improving the transfer performance.

## Transferability map generalizable across subtasks

Considering that the above results verify the effectiveness of the transferability map in measuring similarity across various properties, we next sought to explore whether the transferability map is generalizable across subtasks within these properties. Among the aforementioned three categories of properties, physiology focuses on macroscopic life systems, biophysics uses physical methods to study biological phenomena, while physical chemistry analyzes the principles of the chemical behavior of material systems. Here, we consider knowledge transfer across (1) two different physiology multitask datasets: from Tox21[47] to SIDER[51], and vice versa; (2) one biophysics and one physiology multitask datasets: from MUV[52] to Tox21. Each dataset is associated with several binary classification tasks. We treat each subtask within the target dataset as an individual target dataset and consider the source dataset as a multisource dataset. The pre-training and fine-tuning strategy remains consistent with the main experiment.

The transferability maps between Tox21 and SIDER, as well as between MUV and Tox21, are depicted in Fig. S2 of Supplementary Results. Figure 4 shows the relationship between the PGM distance and the transfer performance in various settings. Specifically, we select three subtasks at regular intervals within each multitask target dataset for analysis. The transfer performance is expected to decrease as the PGM distance increases in all settings. In addition, transferring knowledge within physiology datasets, including Tox21 → SIDER and SIDER → Tox21, consistently shows this trend in over 80% of the target subtasks. Similarly, MUV → Tox21 exhibits the same trend in over 70% of the target subtasks. This observation confirms that knowledge transfer within the same category outperforms transfer between different categories. Overall, it shows that in these cases, the transferability map can be well generalized across the subtasks within the multitask properties.

## Ablation study

To gain deeper insights into the effectiveness of each module in PGM, we conduct ablation studies for subsequent analysis: (1) Whether PGM is in a computation-efficient manner? (2) Does the effectiveness of PGM depend on the target dataset size? (3) Does PGM exhibit any relationship with the differences in dataset sizes across diverse tasks? Accordingly, we carry out three experimental groups: (1) Impact of training epochs of PGM, (2) Impact of target dataset size, (3) Fairness comparison on dataset sizes.

**Impact of training epochs of PGM**. To investigate the impact of the number of reiterative training epochs for computing PGM, we conduct PGM experiments with varying epochs. As shown in Fig. 5, the absolute PGM distance values decrease as the number of training epochs increases

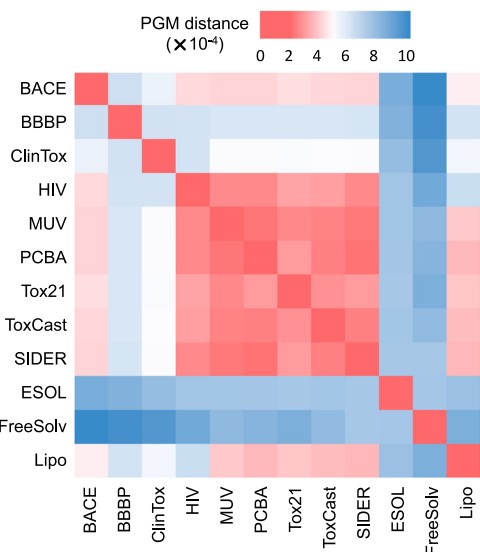

**Fig. 2 | Transferability map illustrating the task-relatedness between the 12 benchmark molecular property prediction datasets.** Red cells with smaller values indicate higher task similarity, while blue ones with larger values indicate lower task similarity.

from 1 to 10 and then to 20. Despite this decrease, the trends regarding task similarity remain consistent across the three maps. PGM perfectly satisfies our need for computation efficiency, as even one epoch of principle gradient suffices. Based on this observation, we empirically recommend applying PGM on these datasets with a few reiterative training epochs (e.g., around 10) to alleviate the abnormal gradients. Due to page limit, we depicte more representative transferability maps at several key epochs and difference heatmaps in Fig. S3 of Supplementary Results.

**Impact of target dataset size.** To study the impact of target dataset size, we evaluate the correlation between the PGM distance and the transfer performance with different target dataset sizes. We employ Kendall's $\tau$ [53] to assess the correlation between two rankings ordered by transfer performance and PGM distances. The target datasets are generated by randomly sampling subsets from a original target dataset with varying percentages, while the remaining 11 datasets consisting the complete dataset are used as source datasets. Figure 6 shows that the PGM distance generally has a significant linear correlation with the transfer performance, particularly when the target subset size exceeds 40% of the target dataset. The absolute values of Kendall's $\tau$ drop less significantly in classification target tasks compared to regression target tasks when the target sample size decreases. This can be attributed to the larger sizes of the classification datasets overall. These findings demonstrate the superior and extensible performance of PGM, serving as a reference standard for transfer learning in molecular property prediction, even with limited target samples.

**Fairness comparison on dataset sizes.** As for the problem that datasets on different properties vary in dataset sizes, we conduct fairness comparison experiments. Considering the availability of dataset sizes (Table S1 in Supplementary Note 2), we randomly select 8000 molecules from each of the source datasets, namely HIV[54], MUV[52], Tox21 (all 7831 molecules included)[47], and ToxCast[55]. We randomly select 1000 molecules from each of the target datasets, including BACE[50], BBBP[48], ClinTox[56], and SIDER[51].

Similarly, in Fig. 7, the PGM distance generally exhibits a significant linear correlation with the transfer performance, consistent with the main experimental finding. This demonstrates the robustness of PGM, as the

correlation between the PGM distance and the transfer performance remains unaffected by variations in data volume across different datasets.

## Conclusion

In this study, we propose Principal Gradient-based Measurement (PGM) to support transferability quantification for molecular property prediction datasets. Specifically, we design a principal gradient to approximate model optimization, which performs on source and target datasets to realize transferability mesaure between datasets. Furthermore, we build a transferability map based on PGM to access task-relatedness prior to applying transfer learning. Both theoretical and empirical studies demonstrate that PGM strongly correlates with the transfer performance of molecular property prediction, making it a quantified transferability measure for source dataset selection.

The superiority in predicting the transferability between molecular property prediction datasets reflects the potential of our PGM framework, and there are interesting and promising future works based on PGM. (1) Extending the one-to-one transfer manner to simultaneously selecting the top-n similar source molecular property prediction datasets during the pre-training phase. (2) Improving the transferability metric to guide the design of neural networks and training objectives for molecule generation and optimization tasks.

## Methods

### Principal Gradient-based Measurement (PGM) for quantifying transferability

**Notation and problem definition.** Given a target task $T_t$ and a predefined set of source tasks $\{(T_{s(n)}\}_{n=1}^N$, the goal of this work is to select the most similar source task for the target task via quantifying their transferability. The corresponding target dataset and source datasets are represented by $D_t$ and $\{(D_{s(n)}\}_{n=1}^N$. The model, denoted by $F = g \cdot w$, comprises a feature extractor $g$ and a predictor $w$. Prior research has demonstrated that higher layers encode more semantic patterns specific to source tasks, while lower-layer features are more generic and related to transferability[57]. Especially when a pair of tasks lack sufficient similarity, transferring higher layers may hurt the performance of a target task. In other words, the transferability is more related to the feature extractor. Thus, we consider the transferability solely focusing on the feature extractor $g$. We denote the model optimization procedure from the initial point to the optimal convergence point in the parameter space, where the optimal feature extractor $g^*$ is initialized from $g_0$. Building on the feature extractor, we define the transferability as the distance between the optimal feature extractor for the source task and that for the target task with the metric function $d(\cdot)$:

**Definition 1. Transferability.** The transferability of a feature extractor $g$ from the source task $T_s$ to the target task $T_t$, denoted by $PGM_{T_s \to T_t}(g)$, is measured by the distance between the optimal feature extractor for the source task $g^*_{T_s}$ and that for the target task $g^*_{T_t}$: $PGM_{T_s \to T_t}(g) = d(g^*_{T_s}, g^*_{T_t})$.

This definition of transferability can be used for selecting a source dataset among $\{D_{s(n)}\}_{n=1}^N$ for a given target dataset $D_t$ in transfer learning.

**PGM.** Based on the feature extractor, the loss function value at the initialized point $g_0$ and the optimal convergence point $g^*$ are denoted by $\mathcal{L}(g_0)$ and $\mathcal{L}(g^*)$, respectively. We first initialize the model with random weight $(g_0, w_0)$ and make a single forward pass of model training on a dataset to compute the gradient of feature extractor parameters, denoted by $\nabla\mathcal{L}(g_0)$. Second, instead of updating model parameters, we reinitialize the model weights to the same starting point and still perform a single forward pass to collect $\nabla\mathcal{L}(g_0)$. We repeat this process multiple times and then compute the expectation of all the collected gradients, so as to mitigate the potential impact of abnormal gradients and ensure robustness. Finally, we name the expectation of the gradients as the principal gradient, defined by

$$PGM(g_0) = E_{g_0}[\nabla\mathcal{L}(g_0)],\qquad(1)$$

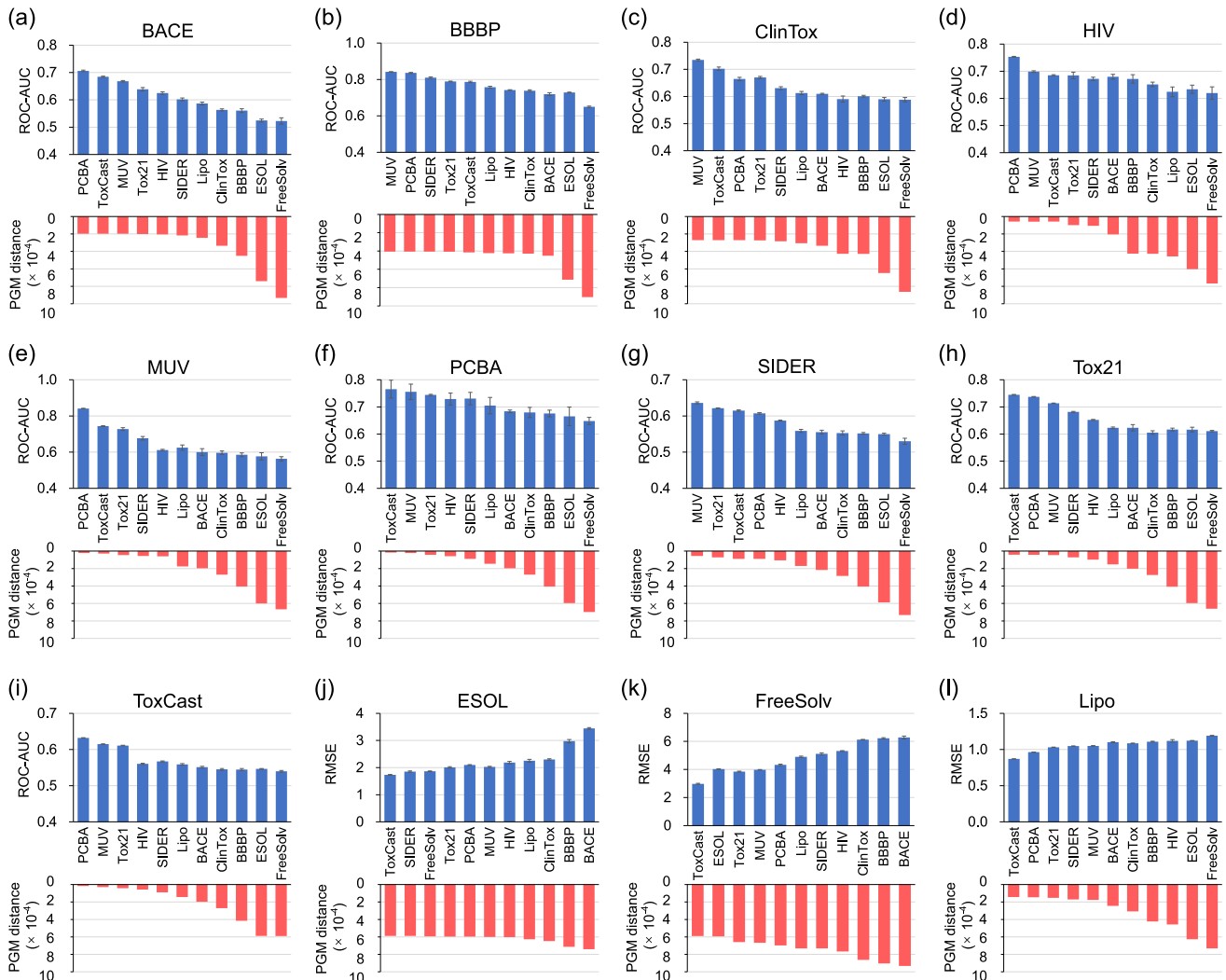

**Fig. 3 | Comparison of the PGM distance and the transfer performance on the 12 target datasets.** The 12 targets include 9 classification datasets (**a–i**) and 3 regression datasets (**j–l**). For transfer learning experiments, the mean and standard deviation values for five experimental runs are reported.

**Transferability quantification by PGM**. Model optimization is aptly characterized as a gradient descent procedure, where the model parameters are iteratively adjusted to minimize the loss function. This process can be mathematically formulated utilizing Taylor's formula[58]. To strike a balance between efficiency and effectiveness, we only employ the first-order gradient. Concretely,

$$\mathcal{L}(g^*) = \mathcal{L}(g_0) + \nabla\mathcal{L}(g_0)(g^* - g_0) + \alpha, \qquad (2)$$

where $\alpha$ represents the remainder beyond the first-order approximation. From (2), we derive

$$(g^* - g_0) + \frac{\alpha}{\nabla\mathcal{L}(g_0)} = \frac{\mathcal{L}(g^*) - \mathcal{L}(g_0)}{\nabla\mathcal{L}(g_0)}, \qquad (3)$$

where $(g^* - g_0)$ denotes the optimization distance between the initial point and the optimal convergence point. However, since the optimal feature extractor $g^*$ is inaccessible without optimization, we need a more simplified and accessible characterization for the distance. As $\alpha$ contains high-order terms, indicating the complexity of the optimization process, we reasonably assume that $(g^* - g_0)$ and $\frac{\alpha}{\nabla\mathcal{L}(g_0)}$ jointly reflect the optimization difficulty of $g$, which is proportional to $\frac{1}{\nabla\mathcal{L}(g_0)}$. Therefore, we define $\frac{1}{\nabla\mathcal{L}(g_0)}$ as a distance

metric to measure the distance from $g_0$ to $g^*$ as follows:

$$d(g_0, g^*) = \frac{1}{\nabla\mathcal{L}(g_0)}, \qquad (4)$$

Next we consider the distance metric on source and target task as

$$\begin{aligned} d_{T_s}(g_0, g^*) &= \frac{1}{\nabla\mathcal{L}_{T_s}(g_0)}, \\ d_{T_t}(g_0, g^*) &= \frac{1}{\nabla\mathcal{L}_{T_t}(g_0)}. \end{aligned} \qquad (5)$$

We define the distance between the optimal feature extractor for the source task $g^*_{T_s}$ and that for the target task $g^*_{T_t}$ by parameter matrix subtraction as

$$\begin{aligned} d(g^*_{T_s}, g^*_{T_t}) &= \| d_{T_s}(g_0, g^*) - d_{T_t}(g_0, g^*) \|_2 \\ &= \| \frac{1}{\nabla\mathcal{L}_{T_s}(g_0)} - \frac{1}{\nabla\mathcal{L}_{T_t}(g_0)} \|_2 \\ &= \frac{\| \nabla\mathcal{L}_{T_t}(g_0) - \nabla\mathcal{L}_{T_s}(g_0) \|_2}{\| \nabla\mathcal{L}_{T_s}(g_0) \nabla\mathcal{L}_{T_t}(g_0) \|_2} \\ &\geq \frac{\| \nabla\mathcal{L}_{T_t}(g_0) - \nabla\mathcal{L}_{T_s}(g_0) \|_2}{\| \nabla\mathcal{L}_{T_s}(g_0) \|_2 \, \| \nabla\mathcal{L}_{T_t}(g_0) \|_2}, \end{aligned} \qquad (6)$$

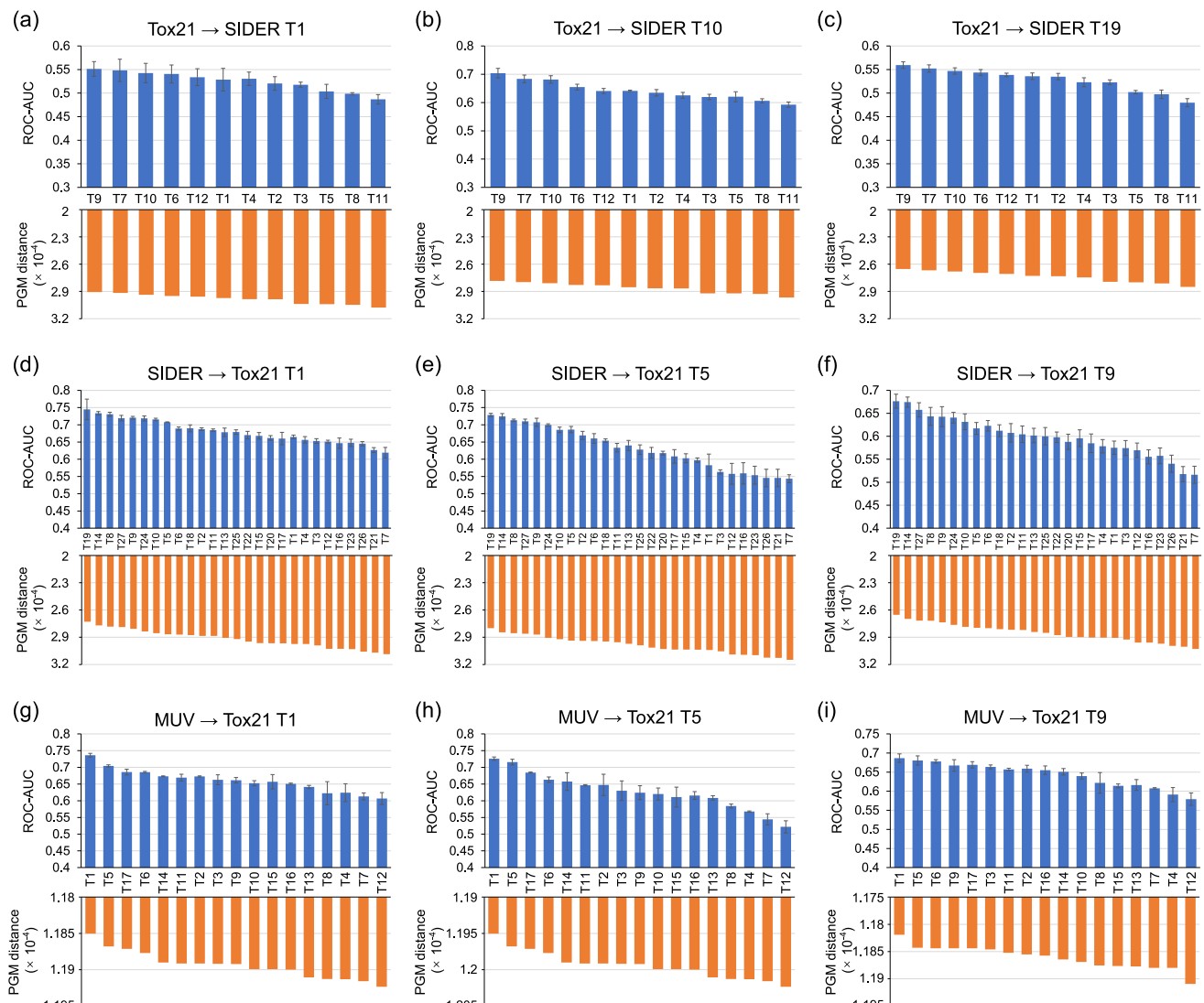

**Fig. 4 | Comparison of the PGM distance and the transfer performance across subtasks within different molecular property prediction datasets.** The comparison performance on (1) two different physiology multitask datasets: from Tox21 to SIDER (**a, b, c**), and vice versa (**d, e, f**); (2) one biophysics and one physiology multitask datasets: from MUV to Tox21 (**g, h, i**). For each multitask target dataset, we select three subtasks at regular intervals within the dataset as individual target datasets. For transfer learning experiments, the mean and standard deviation values for five experimental runs are reported.

where the inequality is derived from Cauchy-Schwarz inequality[59].

As calculating the gradient only once may lead to abnormal value, we apply PGM by computing the expectation of gradients in (6):

$$
\begin{aligned}
PGM_{T_s \rightarrow T_t}(g) &= d(g^*_{T_s}, g^*_{T_t}) \\
&= \frac{\| E_{g_0}[\nabla \mathcal{L}_{T_t}(g_0)] - E_{g_0}[\nabla \mathcal{L}_{T_s}(g_0)] \|_2}{\| E_{g_0}[\nabla \mathcal{L}_{T_s}(g_0)] \|_2 \| E_{g_0}[\nabla \mathcal{L}_{T_t}(g_0)] \|_2},
\end{aligned}
\tag{7}
$$

The distance metric measures the gap between $g^*_{T_s}$ and $g^*_{T_t}$. Finally, we use $d(g^*_{T_s}, g^*_{T_t})$ named PGM distance to denote the transferability from the source task to the target task. The above process of quantifying transferability with PGM is efficient, as the only computation is repeatedly making a single forward pass of model training on datasets without optimization. Further algorithm details are in Supplementary Note 1.

**Transferability map-guided transfer learning**

Based on PGM, we develop a quantitative transferability map to provide a panoramic view of task-relatedness between the source and target molecular property prediction datasets. Specifically, we employ the pairwise PGM distances as the transferability metric between these datasets. In the main

experiment conducted on 12 benchmark datasets from MoleculeNet[42], we utilize the 12 datasets to build the transferability map. In the experiment generalizable across subtasks, we use all the subtasks within corresponding molecular property prediction datasets to build the transferability maps. With the guidance of the transferability maps, we perform transfer learning in various settings and for different tasks, which can be found in Results. Finally, We evaluate the correlation between the predicted transferability by PGM and the transfer performance.

**Experimental setup**

**Datasets**. We use 12 benchmark datasets from MoleculeNet[42], including 9 binary classification tasks and 3 regression tasks. The datasets cover molecular data from a wide range of domains, such as physical chemistry, biophysics, and physiology. In PGM, We conduct experiments on all datasets using the complete dataset. In pre-training and fine-tuning, we perform five independent runs on five random-seeded scaffold splitting for all datasets with a train/validation/test ratio of 8:1:1, as suggested by the MoleculeNet[42]. Scaffold splitting[60] splits molecules based on their scaffolds (molecular substructures), which can better evaluate the generalization ability of the models on out-of-distribution data samples.

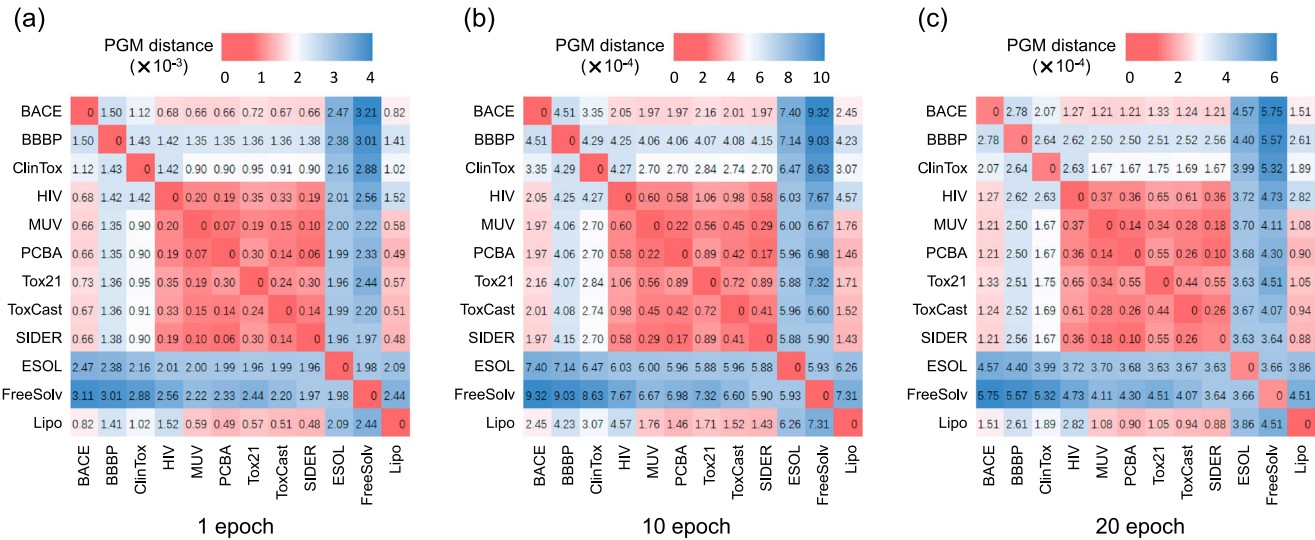

**Fig. 5 | Impact of training epochs of PGM.** Transferability maps of performing PGM with (**a**) 1, (**b**) 10, and (**c**) 20 epochs to explore the minimum number of epochs required for training. Red cells with smaller values indicate higher task similarity, while blue ones with larger values indicate lower task similarity.

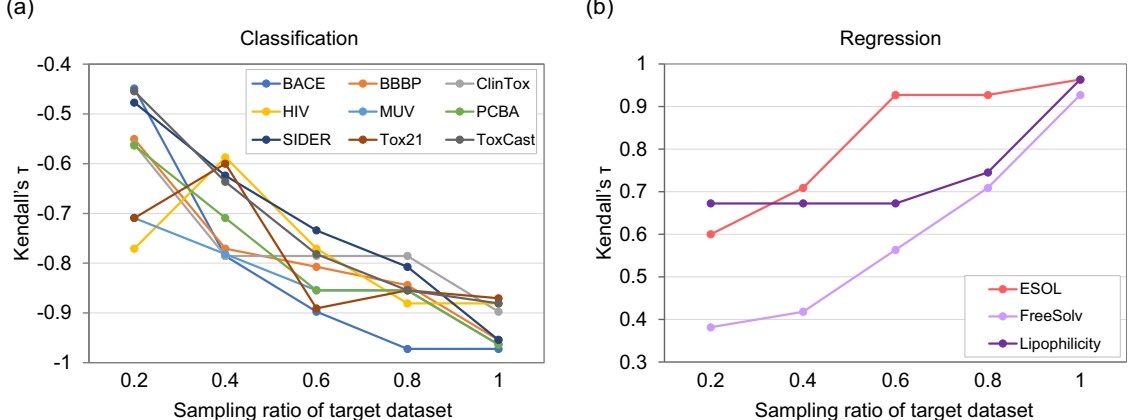

**Fig. 6 | Impact of target dataset size.** Impact of the sampling ratio of target datasets to the performance of PGM, when transferring the remaining 11 source datasets to each target dataset in both **a** classification and **b** regression tasks.

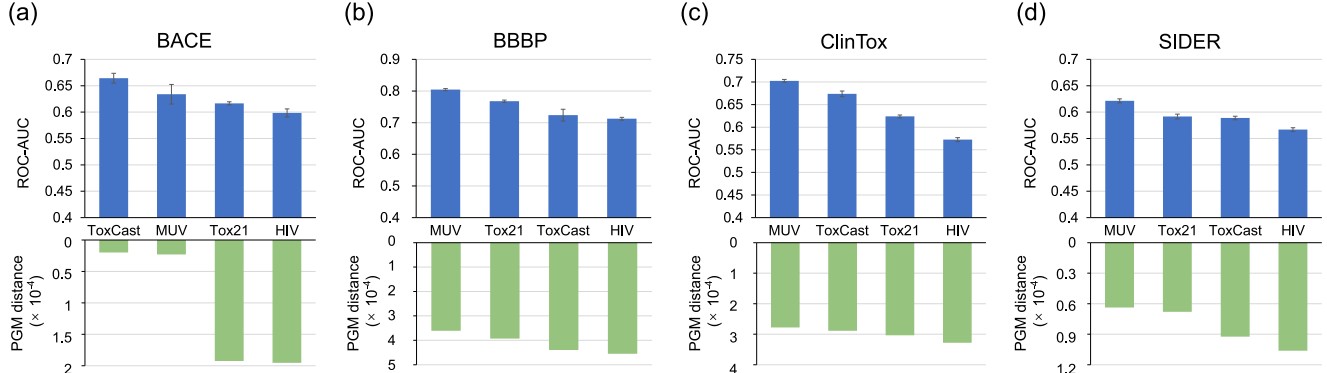

**Fig. 7 | Impact of source and target dataset size to the performance of PGM.** Impact evaluated by comparing the PGM distance and the transfer performance on transferring the four source datasets (HIV, MUV, Tox21, and ToxCast), each containing 8000 molecules, to each of the target datasets (**a** BACE, **b** BBBP, **c** ClinTox, and **d** SIDER), each consisting of 1000 molecules. For transfer learning experiments, the mean and standard deviation values for five experimental runs are reported.

Table S1 in Supplementary Note 2 contains more details about the datasets.

**Implementation details**. We use the open-source chemical analysis tool RDKit[61] to convert raw SMILES into 2D molecular graphs and extract atom features (atom number and chirality tag) and bond features (bond type and bond direction). We adopt graph isomorphism network (GIN)[62] based on DGL-LifeSci package[63] to extract the molecular graph representation in both PGM and transfer learning experiments. GIN is sensitive to the number of layers, and we employ a widely used 3-layer setting. We also use the average pooling as the READOUT function to obtain the graph representation. We adopt a single-layer MLP as the property prediction network.

In PGM, we empirically perform model training for 10 epochs on various datasets. To build the transferability map, we compute pairwise PGM distances between the principal gradient matrices of these datasets. Specifically, we extract the weight parameters from these gradient matrices and concatenate them into a single tensor. Subsequently, we calculate the pairwise PGM distances between these concatenated tensors.

In the transfer learning experiments, during pre-training, we use the Adam optimizer with a learning rate of $1 \times 10^{-3}$, optimizing BCE-WithLogitsLoss for classification and SmoothL1Loss for regression tasks. We train pre-trained models on each dataset with a batch size of 32 and 200 epochs, and select the one with the best performance on the validation metric. This ensures our pre-trained models are well-trained and generalize effectively to new data. In fine-tuning, we initialize the feature extractor from the pre-trained model and train the predictor from scratch on the target task. We fine-tune five times with a batch size of 32 to report the average and standard deviation of performance on the testing set, using ROC-AUC for classification and RMSE for regression tasks.

PGM is implemented utilizing Pytorch and runs on an Ubuntu Server with NVIDIA GeForce RTX 3090Ti graphics processing units.

**Evaluation metrics**. As suggested by the MoleculeNet[42], we use ROC-AUC as the evaluation metric for the binary classification datasets, for which higher is better. With respect to regression datasets, we use RMSE, for which lower is better. We perform five independent runs with five random seeds for each method and report the mean and the standard deviation of the metrics.

In the ablation study, we measure the similarity between two rankings ordered by transfer performance and PGM distances using Kendall's $\tau$[53], also known as Kendall rank correlation coefficient. Kendall's $\tau$ value ranges from -1 to 1, with higher values indicating consistent rankings, lower values implying reversed rankings, and 0 suggesting unrelated rankings.

## Data availability
The datasets used in our paper are publicly available on the MoleculeNet website https://moleculenet.org/datasets-1.

## Code availability
Codes for training the source data and analyzing the results are available on the Zenodo at https://doi.org/10.5281/zenodo.10071500.

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

## Acknowledgements

This work is supported by the Starry Night Science Fund of Zhejiang University Shanghai Institute for Advanced Study (Grant No. SN-ZJU-SIAS-001), Zhejiang Province High-Level Talents Special Support Program "Leading Talent of Technological Innovation of Ten-Thousands Talents Program" (No. 2022R52046), Zhejiang Provincial Science and Technology Project for Public Welfare (LGG22F020007), and Scientific Research Fund of Zhejiang University (No. XY2023020).

## Author contributions

S.Y. proposed the research, conducted experiments, analyzed the data, and wrote the manuscript. L.J. and Z.Z. contributed technical support in conducting experiments. J.S., L.C. and M.S. provided evaluation and suggestions. Z.F. supervised the overall project.

## Competing interests

The authors declare no competing interests.
