## [Peer Review File · Communications Chemistry]

Reviewers' comments:

Reviewer #1 (Remarks to the Author):

The manuscript "Fast and effective molecular property prediction with transferability map" presents a novel approach for molecular property prediction using transfer learning.

The manuscript introduces PGM as a technique to quantify the transferability of molecular property prediction ability between datasets. This optimization-free method calculates a principal gradient to approximate the direction of model optimization, providing a theoretical connection between the principal gradient and model optimization.

It introduces a transferability map to quantify task-relatedness prior to transfer learning. This map helps in selecting the most suitable source dataset for a given target dataset, addressing the issue of negative transfer and enhancing transfer learning performance.

Points to Address:

1. Abstract sentence "We have theoretically analyzed the close connection between the principal gradient and model optimization" seems out of place.

2. Abstract: "This work contributes to more efficient drug discovery by offering a task-relatedness quantification prior to transfer learning and understanding the relationship between molecular properties."

Why just drug discovery? there are many tasks in material science, chemistry, etc. that would benefit from improved transfer learning. Please make this sentence more general (not just specific to drug discovery). Additionally, please cite/discuss this as an example of multiple fields benefitting from good transfer learning: (Tartarus: A Benchmarking Platform for Realistic And Practical Inverse Molecular Desig: <https://arxiv.org/abs/2209.12487>)

3. Line 1-6, Page 1: Again, please make this more general (similar to point 2), a more broad audience would benefit from this work (not just drug discovery) :)

4. "Additionally, some supervised pre-training tasks unrelated to the downstream task of interest can even degrade the downstream performance." Additional review to cite here: Nigam, A. et al. Assigning confidence

to molecular property prediction. *Expert Opin. Drug Discov.* 16, 1009–1023 (2021).

5. Page 2, line 59 "However, these techniques need high computational costs..." why is this the case? A one-line clarification would help readers.

6. Page 2, point 2 of " Our contributions can be summarized as follows:" this reviewer is a bit confused about what you're getting at. It sounds like the authors are saying the transferability map helps in applying transfer learning to molecular property prediction on relevant datasets, but it's not totally clear. Could you clarify this a bit more? Maybe add some examples or a simpler explanation about how this map works in practice and why it's useful for transfer learning. That would really help get the point across more effectively.

7. Results, Para 1: "In this paper, we propose Principal Gradient based Measurement (PGM), a transferability quantification measure for guiding transfer learning in molecular property prediction, which comprises three major components: (1) PGM, (2) transferability map building, (3) application of

transfer learning. An overview of PGM guided transfer learning is shown in Fig. 1.": There's a noticeable issue with the sentence structure that makes it a bit confusing and, frankly, grammatically off. The way it's written now, it seems to suggest that PGM comprises itself, which doesn't make sense. Please improve this paragraph.

8. Page 3, Line 145: "Based on the invisible optimal point..": The reviewer would suggest changing "optimal point" to "a representative point of the dataset". Optimal points are unclear (there is no definition of

these & there are multiple ways of coming up with an optimal point"

Additionally: "computing the transferability is as prohibitively expensive as...." -- good point. Please mention why it is prohibitively expensive (because one needs to traverse the entire database to find non-random representatives).

9. "Markedly, the gradient exhibits minimal variation from one epoch to the next epoch, making itself an attractive measure for model optimization direction." -- So? Why does low variation help capture "task-related characteristics" as the authors indicate in the previous sentence?

This seems like a repeat sentence from earlier paragraphs "Here, we propose a principal gradient that approximately represents the direction for model optimization on a dataset. The technical details and theoretical analysis of PGM are provided in Methods." let's provide some explanation of the technique here (rather than just referring to the methods). Additionally, let's only use "here, we propose..." max 1-3 times in the manuscript.

10. Page 3, line 176 " We utilize PGM distance as the distance metric, providing a detailed description in Methods" some elaboration (1-2 lines) in the text would help, rather than just referring to the methods.

11. Page 3, line 182 "The pre-training and fine-tuning details are shown in Methods" Again, please provide some elaboration. This is the results section of the manuscript & some motivation of the technique should be provided for the upcoming results.

12. Sections "Transferability map-guided transfer learning improves performance" & "Transferability map generalizable across subtasks." are well written! along with a very clear Fig 3

13. Excellent Ablation study

Overall, the authors have done an excellent job, and the manuscript requires only minor revisions :)

Reviewer #2 (Remarks to the Author):

The authors propose a novel principal gradient-based measurement (PGM) method to quantify the transferability from a source property to a target property for transfer learning without requiring model optimization. They use this approach to build a transferability map for 12 datasets (9 classification tasks (including imbalanced data such as MUV) and 3 regression tasks (2x solubility and 1x lipophilicity)). They also fully train each of these datasets and use these as source datasets to transfer learn to each other dataset (by fine-tuning on the target dataset). By comparing the rankings of each set of fine-tuned models to the PGM metrics, the authors show that PGM correlates well with transferability generally. Subsequent ablation studies show that the PGM is invariant to the number of training epochs, the size of the target data set, and differences in the size of the source and target data sets.

The authors' methodology is thorough. For features they use graphs generated from SMILES strings

(RDKit) plus simple 2D molecular features. For the model they use the DGL-LifeSci GIN graph neural network with 3-layers (termed the feature extractor, which is frozen during fine-tuning) and one fully connected layer for prediction (termed the predictor, which is retrained during fine-tuning). Only the feature extractor is used to calculate the PGM, and thus I believe target labels are not required to calculate PGM; an impressive result that ought to be discussed if true. Overall, the code appears concise and easy to read and data available via a previous publication. To my knowledge, all statistical analyses, such as the calculations of Kendall's Tau, are appropriate and valid.

Overall, I believe the results are very impressive and the methods presented novel. Furthermore, I believe the PGM is applicable to an important problem in machine learning generally and of wide interest to the field. The results presented by the authors seem impressive alone, as well as in comparison to current literature on transferability measures, such as LogME and TransRate. However, I believe some further evidence is required to prove this, and a few other details about the work and a more thorough discussion of previous literature are also required. After these moderate changes, I believe the paper is well worthy of publication in CommChem.

My full point-by-point comments are attached below:

Major Comments

1. The authors select 12 tasks for assessing transferability from MoleculeNet (9 classification and 3 regression tasks). However, MoleculeNet contains 17 distinct datasets overall and the authors do not investigate any of the QM datasets (QM7, QM8, or QM9) or PDBbind, which are part of MoleculeNet. This leaves them short on regression tasks. Can the authors comment on why these datasets were omitted? If not, is it possible to include them?

2. Have the fine-tuned models sufficiently converged? Can the authors show this with learning curves (perhaps in the SI)? It is important that the PGM metrics are being compared against fully-converged pre-trained models. In the code, the default number of epochs appears to be 20 for fine-tuning (compared to 200 for pretraining), but this does not appear to be discussed in the text.

3. I think the authors could try to discuss some more of the trends in Figures 2 and 3 in the text. For example, some things I noticed are:

- It's unexpected that Lipo, a regression task, performs well for many classification tasks.
- ESOL and FreeSolv are always bad predictors according to PGM analysis, even with each other, despite predicting the same property (solubility).
- ESOL is one of the worst source datasets for ToxCast predictions, whilst ToxCast is the best source dataset for ESOL.
- ToxCast is frequently a top performing model.
- HIV, MUV, PCBA, Tox21, ToxCast, SIDER are all very similar tasks according to PGM analysis.

4. The authors calculated their PGMs on 1, 10, and 20 epochs to show that the PGM is invariant to the number of training epochs. In my opinion, a test at 100 epochs (or at least with early stopping) would be more interesting. This is also a more typical value compared to transferability measures in computer

vision.

5. The purpose of the PGM metric is to gauge the transferability of one dataset to another in a time-efficient manner. Thus, the authors should report the time taken to compute their PGM metric (with each number of epochs) and compare this to the time taken for fine-tuning. If it takes longer to compute PGM than it does to fine-tune the model, then in practice one could gauge transferability simply by performing transfer learning instead of predicting it beforehand. I suspect this is not the case, and that PGM is faster, but it is important to show this.

6. I believe a more thorough discussion of previous literature for transferability metrics is necessary. For example, metrics such as TransRate and LogMe are not discussed. These are among the best performing metrics already available from the computer vision literature (both of which work for classification and regression tasks and are shown to be computed more rapidly than fine-tuning). Other examples include LEEP and H-Score. I expect that PGM will outperform these transferability metrics, but if the authors wanted to show this beyond doubt they could compute any appropriate metric for a small subset of the tasks included in the paper (maybe 1-2 regression tasks and 1-2 classification tasks). This would put the results of the paper into greater context.

7. A general issue for transferability prediction in chemistry is that target data can be expensive to compute. In lines 400-401, the authors explain how only the feature extractor is used to calculate the PGM. If I understand this correctly, this means that the labels for the target data are not required to calculate PGM; an extremely impressive result that ought to be highlighted if true. For context, TransRate and LogMe both require the target labels. This would be an alternative way for the authors to demonstrate an advantage of PGM over current best performing metrics for transferability prediction (rather than calculating other metrics, as suggested above).

Minor Comments

13-20: The authors list a few uses of transfer learning but stop at biomedicine. It would be good to highlight uses of transfer learning in a chemical domain and expand the depth of the literature review in this rapidly advancing field. A few examples are included.

- Regio/Stereochem: <https://www.nature.com/articles/s41467-020-18671-7>
- Reaction Barriers: <https://pubs.rsc.org/en/content/articlelanding/2023/dd/d3dd00085k>
- NN Potentials: <https://pubs.rsc.org/en/content/articlelanding/2023/cp/d2cp05793j>
- Catalysis: <https://doi.org/10.1039/d1dd00052g>
- Activation Barriers: <https://pubs.acs.org/doi/10.1021/acs.jpcclett.0c00500>

The symmetric presentation of Figure 2 implies that transfer from dataset A to B should be equally as good as B to A, but is this always the case?

337: Can the authors elaborate on their choice of datasets for the fairness comparison on dataset sizes? Only 4 of the 13 datasets are used. Was this just to save time?

549-552: Can the authors state which hyperparameters they tuned? I can't see any other mention of the

hyperparameters and how they were tuned in the main text or in the python code.

Reviewer #3 (Remarks to the Author):

The authors have proposed an efficient Principal Gradient-based Measurement (PGM) for transferring molecular property prediction ability among different tasks and evaluated its performance on multiple combinations of tasks. Overall, PGM is physically sound and seems to be new. I'd recommend publication if the following questions are properly addressed:

- There are many datasets involved in this study, but little information is available about what they are and why they were chosen (if I didn't miss anything). My questions/suggestions include:
 - Give a very concise description about what they are and why they are selected, put them in the ms where deemed most suitable (main text or SI).
 - Explain their potential connection
- The paper is mostly about numerics, offering no or little physical insights. My questions are:
 - The performance for some task pair is better than another pair. Is this even understandable?
 - For the pair of tasks 1 & 2, the PGM distance $d(1,2) = d(2,1)$, but does this imply equal performance no matter what task is used as the base task? I.e., does it make a difference transferring from 1->2 and 2->1?
 - Can this be explained by other means through rough linear correlation, say, based on the relevant fingerprints from the cheminformatics world?
 - Is it possible that PGM may not be necessary at all, in case of perfect explainability by the above-mentioned linear model?
- Regarding the transfer model per se, it's most commonly seen to be applied from one task to another, how about extending it to more tasks, or make a composite model from task 1 & 2, and transfer to task 3?
- One technical concern is about the explosion of the inverse of gradient (see eq. 4): how likely it may happen? And if it happens, how to fix it?

Response To Reviewers

We sincerely appreciate the editor and all the reviewers for their constructive comments and suggestions on our manuscript entitled “Fast and effective molecular property prediction with transferability map” (Manuscript ID: COMMSCHEM-23-0558). In the revised version, we have addressed the concerns of the reviewers, including expanding the literature review, clarifying the motivations, pointing out the limitations of our proposed method, offering tentative explanations for the results, and providing detailed experimental setups, as well as enhancing the quality of writing and expression. We have enclosed a point-by-point response to the reviewers’ comments, with revisions highlighted in blue in the manuscript. Figures and tables in the revised manuscript are denoted as Fig. 1, Table 1, while those in supplementary information are marked as Fig. S1, Table S1. We have also cited more related literature, and the citation numbers in the response follow those in the revised manuscript, e.g. RXNMapper[25]. We hope that these revisions successfully address your concerns.

TO REVIEWER 1

Comment 1: Abstract sentence “We have theoretically analyzed the close connection between the principal gradient and model optimization” seems out of place.

Response 1: Thank you for your suggestion. We agree with you that the term “theoretically analyzed” may not accurately reflect the mathematical proof presented in our paper. We have revised the abstract sentence to provide a more accurate description.

Page 1

“We have analyzed the close connection between the principal gradient and model optimization through mathematical proof.”

Comment 2: a) Abstract: “This work contributes to more efficient drug discovery by offering a task-relatedness quantification prior to transfer learning and understanding the relationship between molecular properties.” Why just drug discovery? there are many tasks in material science, chemistry, etc. that would benefit from improved transfer learning. Please make this sentence more general (not just specific to drug discovery). **b)** Additionally, please cite/discuss this as an example of multiple fields benefitting from good transfer learning: (Tartarus: A Benchmarking Platform for Realistic And Practical Inverse Molecular Desig: <https://arxiv.org/abs/2209.12487>).

Response 2: a) Thank you for your valuable suggestion. As you rightly mentioned, efficient transfer learning will benefit various fields, such as designing drugs, catalysts, and materials. We have revised the abstract sentence to reflect a broader perspective, acknowledging the potential impact on various fields such as material science and chemistry.

b) We also cite and discuss the latest work on molecular design benchmarks named Tartarus, which develops realistic and practical benchmarks reflecting the complexity of molecular design for real-world applications. It will be also benefit from good transfer learning. Thank you very much for sharing this paper with us.

Page 1

“This work contributes to more efficient discovery of drugs, materials, and catalysts by offering a task-relatedness quantification prior to transfer learning and understanding the relationship between chemical properties.”

Page 1, Line 6-12

“In recent years, artificial intelligence (AI) technologies have come mainstream in this area, and AI-guided chemical design can efficiently explore chemical space while improving performance based on experimental feedback, showing promise from laboratory research to real-world industry applications[3].”

Comment 3: 3. Line 1-6, Page 1: Again, please make this more general (similar to point 2), a more broad audience would benefit from this work (not just drug discovery) :).

Response 3: Thank you very much for your suggestion. We have revised the sentence from a broader perspective.

Page 1, Line 1-6

“Molecular property prediction, which involves identifying molecules with desired properties[1,2], poses a critical challenge prevalent across various scientific fields. It holds particular significance in chemistry for designing drugs, catalysts, and materials.”

Comment 4: “Additionally, some supervised pre-training tasks unrelated to the downstream task of interest can even degrade the downstream performance.” Additional review to cite here: Nigam, A. et al. Assigning confidence to molecular property prediction. Expert Opin. Drug Discov. 16, 1009–1023 (2021).

Response 4: Thank you for sharing the valuable work on uncertainties in molecular property prediction. We appreciate the mention in the article about dataset uncertainty on potential biases and dataset size, which requires more research when applying transfer learning to these datasets. We have cited the paper in the introduction.

Page2, Line 43-46

“Additionally, some supervised pre-training tasks unrelated to the downstream task of interest can even degrade the downstream performance[23,28].”

Comment 5: Page 2, line 59 “However, these techniques need high computational costs...” why is this the case? A one-line clarification would help readers.

Response 5: Thank you for reminding us that here needs clarification. These representation similarity measurements build a similarity graph between examples for each task based on representations by a pre-trained model on this task, and take the graph similarity across tasks as the similarity between biological compounds. As they stand in need of model optimization across all datasets, their computational costs are as high as fine-tuning with target tasks. We have clarified the description in our latest manuscript.

Page 2, Line 72-79

“These techniques leverage pre-trained model representations to assess biological compound similarity across different tasks. As they stand in need of model optimization across all datasets, and their computational costs that are as high as fine-tuning with target tasks exclude their applications in quantifying transferability prior to fine-tuning.”

Comment 6: a) Page 2, point 2 of “Our contributions can be summarized as follows:” this reviewer is a bit confused about what you’re getting at. It sounds like the authors are saying the transferability map helps in applying transfer learning to molecular property prediction on relevant datasets, but it’s not totally clear. Could you clarify this a bit more? **b)** Maybe add some examples or a simpler explanation about how this map works in practice and why it’s useful for transfer learning. That would really help get the point across more effectively.

Response 6: a) We sincerely apologize for the confusion caused here. To clarify, the transferability map visualizes inter-property correlations, measured by performing PGM on various molecular property prediction datasets. We have revised this point with a simpler explanation in the introduction.

b) For example, in Fig. 1b, a smaller distance between the source property A (colored in orange) and the target property (colored in red) indicates a higher level of task similarity. Therefore, for the target property in transfer learning, we would choose property A as the source property dataset rather than property B. We added the example in Results on Page3, Line 160-163.

Page 2, Line 119-125

“Furthermore, we perform PGM on molecular benchmarks to build a transferability map that quantifies inter-property correlations. The map is extensible and can be a reference standard for transfer learning in molecular property prediction, even when applied to a few target samples.”

Comment 7: Results, Para 1: “In this paper, we propose Principal Gradient based Measurement (PGM), a transferability quantification measure for guiding transfer learning in molecular property prediction, which comprises three major components: (1) PGM, (2) transferability map building, (3) application of transfer learning. An overview of PGM guided transfer learning is shown in Fig. 1.”: There’s a noticeable issue with the sentence structure that makes it a bit confusing and, frankly, grammatically off. The way it’s written now, it seems to suggest that PGM comprises itself, which doesn’t make sense. Please improve this paragraph.

Response 7: Thank you for your advice. We have improved this paragraph on Page 3, Line 139-145 as follows:

In this paper, we introduce Principal Gradient-based Measurement (PGM) to quantify transferability between the source and target molecular properties. The framework of PGM guided transfer learning comprises three components as illustrated in Fig.1: (1) PGM, (2) transferability map building, and (3) application of transfer learning.

Comment 8: a) Page 3, Line 145: “Based on the invisible optimal point.”: The reviewer would suggest changing “optimal point” to “a representative point of the dataset”. Optimal points are unclear (there is no definition of these & there are multiple ways of coming up with an optimal point”. **b)** Additionally: “computing the transferability is as prohibitively expensive as...” – good point. Please mention why it is prohibitively expensive (because one needs to traverse the entire database to find non-random representatives).

Response 8: a) We apologize for not making the term “optimal point” clear in the manuscript. We understand the importance of using precise terminology, especially when it pertains to concepts central to machine learning. In light of your feedback, we would like to change “optimal point” to “optimal convergence point” and provide a clear definition. Optimal convergence point refers to a position within the model’s parameter space where the model achieves the highest performance or lowest loss, as per the criteria established through training. The model optimization process aims to move from the initial point towards the optimal convergence point by iteratively updating the parameters to reduce the loss function. In many learning algorithms, this is achieved through methods like gradient descent. This definition both aligns with machine learning terminology and your notion “a representative point of the dataset”. We hope this definition addresses your concerns, and we have revised the manuscript accordingly.

b) Regarding the second point about the computational cost, we have provided clarification. The computational cost is attributed to identifying the optimal convergence points, which needs to train and evaluate models on multiple source datasets and the target dataset. As mentioned above regarding the model optimization, it involves iteratively updating model parameters, which requires significant computational power and time. When assessing transferability, our proposed method is specifically designed to be optimization-free, thereby bypassing the need for this computationally expensive model training and iterative parameter updating process. In our latest manuscript, we have explained that the prohibitive expense is due to the model optimization process.

Page 3, Line 146-158:

“The model optimization process aims to move from the initial point towards the optimal convergence point (Fig. 1a), where the model’s parameters minimize the loss function and the model achieves peak performance. This optimization process involves iteratively

updating the model parameters to reduce the loss function, commonly employing gradient descent as the preferred method. Building upon this conceptualization, the model is initialized from the same initial point and subsequently optimized on both source and target molecular property prediction datasets.”

Page 3, Line 165:

“Based on the invisible optimal convergence point, computing the transferability by optimizing models is as prohibitively expensive as training models on the given target dataset and all alternative source datasets, while the transferability offers benefits only when it can be calculated as a prior.”

Comment 9: a) “Markedly, the gradient exhibits minimal variation from one epoch to the next epoch, making itself an attractive measure for model optimization direction.” – So? Why does low variation help capture “task-related characteristics” as the authors indicate in the previous sentence? This seems like a repeat sentence from earlier paragraphs. **b)** “Here, we propose a principal gradient that approximately represents the direction for model optimization on a dataset. The technical details and theoretical analysis of PGM are provided in Methods.” let’s provide some explanation of the technique here (rather than just referring to the methods). **c)** Additionally, let’s only use “here, we propose...” max 1-3 times in the manuscript.

Response 9: a) We sincerely apologize for the oversight statement here. Actually, it is the gradients, rather than the gradient’s minimal variation, that can capture the relationship with task-relatedness. We will introduce several recent studies that explore the relationship between gradients and task-relatedness. For example, Grad2Task posits that gradients can be used as features to capture task nature and distinguish between tasks under a meta-learning framework. Additionally, in multitask learning, gradients from different tasks may conflict with one another, a phenomenon known as gradient conflict. Researchers have proposed gradient surgery techniques, such as PCGrad and GradNorm, to harmonize task gradients and enhance multitask performance. Based on these studies, we propose PGM based on gradients to quantify the transferability concerning task-relatedness. We have revised the manuscript accordingly.

b) We have included a concise explanation of the PGM technique in Results as follows: The proposed principal gradient, obtained through model re-initialization and gradient expectation calculation, approximately indicates the direction for model optimization on a dataset. Afterwards, PGM measures the transferability as the distance between the principal gradient obtained from model training on the source dataset and that derived from the target dataset.

c) We have make necessary revisions to limit the use of “here, we propose...” within 3 times in the latest manuscript.

Page 3, Line 172-184

“Gradients can capture intrinsic task-related characteristics, playing a strong predictive role in model optimization on a dataset, as evidenced by recent studies[43-46]. Grad2Task[43] posits that gradients can be used as features to capture task nature and distinguish between tasks under a meta-learning framework. Numerous gradient surgery techniques[44,45] have been proposed to address gradient conflict and improve model performance in multitask learning. In light of gradients being explored in the context of task-relatedness, we derive PGM, a simple measure based on gradients, to quantify the transferability concerning task-relatedness.”

Page 3, Line 184-192

“Specifically, the proposed principal gradient, obtained through model re-initialization and gradient expectation calculation, approximately indicates the direction for model optimization on a dataset. Afterwards, PGM measures the transferability as the distance between the principal gradient obtained from model training on the source dataset and that derived from the target dataset.”

Comment 10: Page 3, line 176 “We utilize PGM distance as the distance metric, providing a detailed description in Methods” some elaboration (1-2 lines) in the text would help, rather than just referring to the methods.

Response 10: Thanks for the suggestion. We have provided some elaboration of the method as follows: We utilize the PGM distance as the distance metric, which is measured by the difference between the principal gradients of model training on each pair of the datasets.

Page 4, Line 202-206

“We utilize the PGM distance as the distance metric, which is measured by the difference between the principal gradients of model training on each pair of the datasets.”

Comment 11: Page 3, line 182 “The pre-training and fine-tuning details are shown in Methods” Again, please provide some elaboration. This is the results section of the manuscript & some motivation of the technique should be provided for the upcoming results.

Response 11: Thanks for the comments. We have provided some elaboration of the pre-training and fine-tuning details as follows: During the fine-tuning phase, we fix the feature extractor initialized from the pre-trained model, while train the predictor from scratch on the target task.

Page 4, Line 211-214

“During the fine-tuning phase, we fix the feature extractor initialized from the pre-trained model, while train the predictor from scratch on the target task.”

Comment 12: Sections “Transferability map-guided transfer learning improves performance” & “Transferability map generalizable across subtasks.” are well written! along with a very clear Fig 3.

Response 12: We sincerely appreciate the reviewer for these positive comments on our work.

Comment 13: Excellent Ablation study.

Response 13: We are sincerely grateful to the reviewer for the positive feedback on our work.

TO REVIEWER 2

Major Comments:

Comment 1: The authors select 12 tasks for assessing transferability from MoleculeNet (9 classification and 3 regression tasks). However, MoleculeNet contains 17 distinct datasets overall and the authors do not investigate any of the QM datasets (QM7, QM8, or QM9) or PDBbind, which are part of MoleculeNet. This leaves them short on regression tasks. Can the authors comment on why these datasets were omitted? If not, is it possible to include them?

Response 1: Thank you for your thoughtful comments on the selection of datasets. In our study, we chose to focus on a subset of MoleculeNet datasets that best align with our research objectives and the capabilities of our model. MoleculeNet provides a diverse range of datasets across quantum mechanics, physical chemistry, biophysics, and physiology. However, as indicated by Huang et al., there is a significant difference in task-relatedness between quantum chemistry datasets and those of other categories. Molecules in quantum chemistry have a wide array of both real and hypothetical structures to facilitate the exploration of functional groups, while molecules in other categories are more focused on structure to predict and optimize properties for drug design. Moreover, Nigam et al. have pointed out that a significant bias in training data could cause models only to learn the inherent bias rather than physically meaningful relationships. We aimed to select datasets where such biases are minimized to ensure our model captures relevant chemical properties. Regarding PDBbind, this dataset involves protein-ligand interactions, which require specific featurization techniques that are not directly supported by our GIN framework. Similarly, the QM datasets are also not readily compatible with our framework. Using feature extractor in different frameworks will lead to uneven performance gain, even for the same pair of source and target tasks. To maintain a consistent feature extractor framework and control for variables, we did not incorporate QM and PDBbind in our current analysis. We acknowledge that these datasets are important, and we aim to generalize our findings and methods to a wider range of AIDD tasks and datasets, including them in future work, as noted in our manuscript.

[1] Huang, T. et al. Learning to Group Auxiliary Datasets for Molecule. *Advances in Neural Information Processing Systems* 36 (2024).

[2] Nigam, A. et al. Assigning confidence to molecular property prediction. *Expert opinion on drug discovery* 16 (9), 1009–1023 (2021).

Comment 2: a) Have the fine-tuned models sufficiently converged? Can the authors show this with learning curves (perhaps in the SI)?
b) It is important that the PGM metrics are being compared against fully-converged pre-trained models. In the code, the default number of epochs appears to be 20 for fine-tuning (compared to 200 for pretraining), but this does not appear to be discussed in the text.

Response 2: a) Thank you for your comments regarding the convergence of fine-tuned models. We have included learning curves as Fig. S1 in the Supplementary Information. These curves illustrate that convergence is typically achieved between 10 to 20 epochs across almost all datasets. To avoid overfitting, we intentionally set the fine-tuning epochs to 20 on these datasets.

b) Concerning the pre-trained models, our experiments show convergence on all datasets typically occurs between 100 to 200 epochs. We ensure the convergence of the pre-trained models by training them up to 200 epochs. We then select the model that achieves the highest performance based on the validation metric. This strategy ensures we choose a model that is both converged and optimally performs on unseen data. The particulars of this process are elucidated in *Methods*.

Page 11, Line 592-597

“We train pre-trained models on each dataset with a batch size of 32 and 200 epochs, and select the one with the best performance on the validation metric. This ensures our pre-trained models are well-trained and generalize effectively to new data.”

Comment 3: I think the authors could try to discuss some more of the trends in Figures 2 and 3 in the text. For example, some things I noticed are:

- a) It's unexpected that Lipo, a regression task, performs well for many classification tasks.
- b) ESOL and FreeSolv are always bad predictors according to PGM analysis, even with each other, despite predicting the same property (solubility).
- c) ESOL is one of the worst source datasets for ToxCast predictions, whilst ToxCast is the best source dataset for ESOL.
- d) ToxCast is frequently a top performing model.
- e) HIV, MUV, PCBA, Tox21, ToxCast, SIDER are all very similar tasks according to PGM analysis.

Response 3: Thank you for your valuable suggestions. We have tried to discuss some trends in Figure 2 and 3 based on the current literature and our understanding. We offer explanations where feasible and recognize that some patterns merit further investigation. In response to the examples you've noticed, we provide the following tentative explanations:

- a) Upon reviewing Figures 2 and 3, it appears that Lipo exhibits moderate predictive performance for target classification tasks in SIDER, Toxcast, Tox21, PCBA, and MUV. This suggests that the features learned from Lipo may capture some underlying patterns that are beneficial for these datasets.
- b) ESOL and FreeSolv pose challenges as predictors, potentially owing to biases inherent to their limited sizes, which may elevate the risk of overfitting.
- c) ESOL's underperformance may attribute to its limited size, as I mentioned in b). While the potential benefits of ToxCast for ESOL merit further investigation.
- d) Since ToxCast, Tox21, and ClinTox are all toxicity-related, this commonality likely aids ToxCast's effective transfer to both datasets. The specific qualities that enable ToxCast's broad applicability merit further investigation.
- e) The observed similarity among HIV, MUV, and PCBA may be attributed to their large size and useful out-of-distribution information, which enhances the generalizability of predictive models. This factor may contribute to their alignment with Tox21, ToxCast, and SIDER in our analysis. Tox21 and ToxCast are inherently related due to their focus on toxicity, explaining their close grouping. As for SIDER, its connection to the group is likely because it encompasses information on drug side effects and toxicity.

Fig. S1: Training and validation curves of fine-tuning on different target datasets. The solid and dashed lines indicate the training and validation curves, respectively.

Page 5, Line 274-282

“Another interesting finding is that, among all source datasets, PCBA is a top-three performer in boosting the performance of almost all target datasets in classification scenarios, due to its diverse structure or useful out-of-distribution information. Additionally, despite belonging to distinct domains, some datasets exhibit higher task similarity, such as HIV, MUV, PCBA, Tox21, ToxCast, and SIDER.”

Comment 4: The authors calculated their PGMs on 1, 10, and 20 epochs to show that the PGM is invariant to the number of training epochs. In my opinion, a test at 100 epochs (or at least with early stopping) would be more interesting. This is also a more typical value compared to transferability measures in computer vision.

Response 4: Thank you for suggesting further analysis on the stability of PGM over an extended range of training epochs. In response, we have included representative transferability maps at epochs 1, 10, 20, 50, 100, and 200, and generated difference heatmaps for selected epoch comparisons (10 vs. 20, 50 vs. 100, and 100 vs. 200), as shown in Fig. S3. The absolute PGM distance values decrease with more epochs, while the trends in terms of task similarity remain consistent across all nine heatmaps. This consistency confirms the robustness of PGM as a transferability measurement for molecular property prediction. Based on these findings, we maintain our recommendation of using 10 epochs for PGM computation on these datasets, as it provides a balance between computational efficiency and capturing model optimization direction. We have added these findings to the revised Supplementary Information.

Fig. S3: Transferability maps of performing PGM across epochs (a) 1, (b) 10, (c) 20, (d) 50, (e) 100, and (f) 200, and difference heatmaps for selected epoch intervals (g) 10 vs. 20, (h) 50 vs. 100, and (i) 100 vs. 200. Red cells with smaller values indicate closer PGM distances, while blue ones with larger values indicate longer PGM distances.

Comment 5: The purpose of the PGM metric is to gauge the transferability of one dataset to another in a time-efficient manner. Thus, the authors should report the time taken to compute their PGM metric (with each number of epochs) and compare this to the time taken for fine-tuning. If it takes longer to compute PGM than it does to fine-tune the model, then in practice one could gauge transferability simply by performing transfer learning instead of predicting it beforehand. I suspect this is not the case, and that PGM is faster, but it is important to show this.

Response 5: Thanks for the suggestion. We compared the running times of PGM and fine-tuning for three target datasets—BACE, Tox21, and ESOL—randomly selected from each of the three categories as detailed in Table S1. We run the experiments on a server with 2 Intel Xeon Platinum 8180 2.50GHz CPUs and a single NVIDIA GeForce RTX 3090 GPU. We present the training time in Table S2 in the Supplementary Information. Table S2 details the time for transferability quantification (including the time for computing principal gradient at 1, 10, and 20 epochs, as well as the time for computing PGM distance) and the time for fine-tuning. Notably, fine-tuning demands 3 to 16 times more time than transferability quantification for an equivalent 20-epoch period. Furthermore, the transferability quantification outcomes can serve as a reference standard for transfer learning across these datasets, negating the need for repeated recalculations. In contrast, fine-tuning requires re-computation for different scenarios. This highlights the computational efficiency of PGM.

Table S2: Training time (seconds) for different target datasets, utilizing the remaining 11 datasets as sources for each target.

Target Dataset	BACE	Tox21	ESOL
PGM (1epoch)		3.94	
PGM (10epoch)		26.28	
PGM (20epoch)		50.63	
PGM distance	1.58	1.65	1.54
fine-tuning (20epoch)	248.81	825.52	160.16

Comment 6: I believe a more thorough discussion of previous literature for transferability metrics is necessary. For example, metrics such as TransRate and LogMe are not discussed. These are among the best performing metrics already available from the computer vision literature (both of which work for classification and regression tasks and are shown to be computed more rapidly than fine-tuning). Other examples include LEEP and H-Score. I expect that PGM will outperform these transferability metrics, but if the authors wanted to show this beyond doubt they could compute any appropriate metric for a small subset of the tasks included in the paper (maybe 1-2 regression tasks and 1-2 classification tasks). This would put the results of the paper into greater context.

Response 6: Thank you for sharing with us the cutting-edge research on transferability metrics within the field of computer vision, such as TransRate and LogMe. We have now expanded the literature review in our manuscript to include a discussion on transferability metrics as you suggested. We acknowledge these metrics’ significance in model selection and their rapid computation advantage. Our manuscript focuses on leveraging the relatedness of molecular property prediction tasks to bolster transfer learning, a direction which is slightly different from the primary aim of the mentioned metrics. We appreciate your suggestion to assess our work against generalized transferability metrics from computer vision. We plan to explore the applicability of such metrics in AI for pharmaceuticals in subsequent research. We believe the revised literature review provides better context and aligns more closely with our research theme. The revised literature review on Page 2 is as follows:

Page 2, Line 47-57

“Negative transfer primarily stems from suboptimal model and layer choices, as well as insufficient task relatedness, highlighting the need to evaluate transferability prior to applying transfer learning. In computer vision, some researchers have recently focused on selecting the best model from a pool of options by estimating the transferability of each model[29-32]. In molecular property prediction, recent efforts involve investigating the relatedness of the source task to the target task.”

Comment 7: A general issue for transferability prediction in chemistry is that target data can be expensive to compute. In lines 400-401, the authors explain how only the feature extractor is used to calculate the PGM. If I understand this correctly, this means that the labels for the target data are not required to calculate PGM; an extremely impressive result that ought to be highlighted if true. For context, TransRate and LogMe both require the target labels. This would be an alternative way for the authors to demonstrate an advantage of PGM over current best performing metrics for transferability prediction (rather than calculating other metrics, as suggested above)

Response 7: We sincerely apologize for the confusion caused here. To clarify, while PGM focuses on the gradients of the feature extractor to assess transferability, the computation does require target labels. This is because gradients are derived from the backpropagation process, which necessitates label information. The lines 441-449 reference existing research which suggests that transferability is more related to lower layers (the feature extractor) than to higher layers (such as fully connected layers) which tend to be more task-specific. Consequently, in calculating PGM, we emphasize the gradients of the feature extractor, as they are more indicative of the model’s transferability potential. We hope this explanation clarifies the process and addresses your concern.

Minor Comments:

Comment 8: 13-20: The authors list a few uses of transfer learning but stop at biomedicine. It would be good to highlight uses of transfer learning in a chemical domain and expand the depth of the literature review in this rapidly advancing field. A few examples are included.

- Regio/Stereochem: <https://www.nature.com/articles/s41467-020-18671-7>
- Reaction Barriers: <https://pubs.rsc.org/en/content/articlelanding/2023/dd/d3dd00085k>
- NN Potentials: <https://pubs.rsc.org/en/content/articlelanding/2023/cp/d2cp05793j>
- Catalysis: <https://doi.org/10.1039/d1dd00052g>
- Activation Barriers: <https://pubs.acs.org/doi/10.1021/acs.jpcclett.0c00500>

Response 8: Thank you very much for sharing these valuable papers to us. We have incorporated the relevant descriptions in our latest manuscript, and highlight the uses of transfer learning in chemistry.

Page 1, Line 21-27

“In chemistry, transfer learning leverages pre-trained models on extensive or related datasets to facilitate efficient exploration of vast chemical space[13,14] for various downstream tasks. It has been used to predict properties[15,16], plan synthesis[17,18], and explore the space of chemical reactions[19-22].”

Comment 9: The symmetric presentation of Figure 2 implies that transfer from dataset A to B should be equally as good as B to A, but is this always the case?

Response 9: Thank you for your insightful comment. The symmetric presentation of Figure 2 reflects relative degrees of transferability rather than symmetric impact between molecule datasets, as I mentioned in Line 198-202: “When assessing the transferability from potential source datasets to a given target dataset, the distance between source and target datasets serves as an indicator of their knowledge transfer ability.” It is meaningful primarily when comparing potential sources for the same target dataset, or potential targets for the same source dataset. To clarify, when considering a target dataset A and source datasets B and C, if the PGM distance $d(B,A) > d(C,A)$, it indicates that source C is more related to target A. In other words, the transferability $C \rightarrow A$ is higher than $B \rightarrow A$. As demonstrated in Fig.3, for each target dataset, we compare the transferability from various source datasets to the target. Taking computer vision tasks as an example, transferring from object detection to semantic segmentation often yields better results compared to transferring from image classification, because object detection shares more fine-grained spatial and structural similarities with semantic segmentation.

Comment 10: 337: Can the authors elaborate on their choice of datasets for the fairness comparison on dataset sizes? Only 4 of the 13 datasets are used. Was this just to save time?

Response 10: Thank you for your feedback. The selection of datasets for the fairness comparison is based on a careful consideration of dataset sizes. The 12 datasets include 9 classification datasets and 3 regression datasets. Here we focus on the 9 classification datasets, among which 4 have a sample size ranging from 1000 to 2000, while the remaining 5 have approximately 8000 or more samples. Then we randomly select 1000 samples from each of the first 4 datasets to create the target datasets, and similarly, we randomly selected 8000 samples from each of the 4 datasets among the latter 5 to create the source datasets. We also take into account the balance of positive and negative samples in our selection process. We believe that this approach allows us to effectively demonstrate the impact on dataset sizes. We have include a concise explanation of the dataset selection in the latest manuscript. Table S1 contain the details about the datasets, including dataset size.

Page 6, Line 387-388

“Considering the availability of dataset sizes (Table S1),”

Comment 11: 549-552: Can the authors state which hyperparameters they tuned? I can’t see any other mention of the hyperparameters and how they were tuned in the main text or in the python code.

Response 11: We sincerely apologize for the oversight statement here. Actually we used the default parameter settings for the algorithms employed, primarily to save on training time. While it is acknowledged that hyperparameter tuning can potentially improve the transfer learning performance, we believe that the overall trend of results would remain consistent with the findings reported. We have revised the statement in our latest manuscript.

Page 11, Line 592-597

“We train pre-trained models on each dataset with a batch size of 32 and 200 epochs, and select the one with the best performance on the validation metric. This ensures our pre-trained models are well-trained and generalize effectively to new data.”

TO REVIEWER 3

Comment 1: There are many datasets involved in this study, but little information is available about what they are and why they were chosen (if I didn’t miss anything). My questions/suggestions include: **a)** Give a very concise description about what they are and why they are selected, put them in the ms where deemed most suitable (main text or SI). **b)** Explain their potential connection.

Response 1: a) Thank you for your suggestions regarding the datasets. The datasets used in our paper are sourced from the MoleculeNet benchmark, encompassing a wide range of molecular properties, such as physical chemistry, biophysics, and physiology. This diversity is crucial for our study on transfer learning and transferability quantification among molecular properties, as it allows us to comprehensively evaluate the effectiveness of our method. More information of the datasets is detailed concisely in Table S1 in the Supplementary Information.

b) These datasets are categorized into physiology, biophysics, and physical chemistry, with their potential connections as follows: **Biophysics** BACE has been established to gather compounds that could act as the inhibitors of human β -secretase 1 (BACE-1). HIV is from the Drug Therapeutics Program (DTP) AIDS Antiviral Screen, and it aims at predicting inhibit HIV replication. PCBA is a dataset that comprises the biological activities of small molecules generated through high-throughput screening methodologies. Maximum Unbiased Validation (MUV) is another sub-database from PCBA, and is obtained by applying a refined nearest neighbor analysis.

Physiology The Blood-Brain Barrier Penetration (BBBP) dataset measures whether a molecule will penetrate the central nervous system. All three datasets, Tox21, ToxCast, and ClinTox are related to the toxicity of molecular compounds. The Side Effect Resource (SIDER) dataset stores the adverse drug reactions on a marketed drug database.

Physical Chemistry ESOL measures aqueous solubility of common organic small molecules. FreeSolv measures hydration free energy of small molecules in water, which is obtained through molecular dynamics simulations. Lipophilicity is a subset of ChEMBL measuring the molecule octanol/water distribution coefficient.

Supplementary Information, Page 1-2

“Table S1 provides an overview of the molecular datasets used in our paper from MoleculeNet[1]. They are categorized into physiology, biophysics, and physical chemistry. The detailed information for the datasets is listed as follows:

Table S1: Summary of all the benchmarks for molecular property predictions used in this work.

Task Type	Metric	Category	Dataset	# Tasks	# Compounds
Classification	ROC-AUC	Biophysics	BACE	1	1,513
			HIV	1	41,127
			MUV	17	93,087
			PCBA	128	437,997
		Physiology	BBBP	1	2,039
			ClinTox	2	1,478
			SIDER	27	1,427
			Tox21	12	7,831
ToxCast	617		8,575		
Regression	RMSE	Physical chemistry	ESOL	1	1,128
			FreeSolv	1	642
			Lipophilicity	1	4,200

Biophysics BACE[2] has been established to gather compounds that could act as the inhibitors of human β -secretase 1 (BACE-1). HIV[3] is from the Drug Therapeutics Program (DTP) AIDS Antiviral Screen, and it aims at predicting inhibit HIV replication. PCBA[4] is a dataset that comprises the biological activities of small molecules generated through high-throughput screening methodologies. Maximum Unbiased Validation (MUV)[5] is another sub-database from PCBA, and is obtained by applying a refined nearest neighbor analysis.

Physiology The Blood-Brain Barrier Penetration (BBBP)[6] dataset measures whether a molecule will penetrate the central nervous system. All three datasets, Tox21[7], ToxCast[8], and ClinTox[9] are related to the toxicity of molecular compounds. The Side Effect Resource (SIDER)[10] dataset stores the adverse drug reactions on a marketed drug database.

Physical Chemistry ESOL[11] measures aqueous solubility of common organic small molecules. FreeSolv[12] measures hydration free energy of small molecules in water, which is obtained through molecular dynamics simulations. Lipophilicity is a subset of ChEMBL[13] measuring the molecule octanol/water distribution coefficient.”

Comment 2: The paper is mostly about numerics, offering no or little physical insights. My questions are:

- The performance for some task pair is better than another pair. Is this even understandable?
- For the pair of tasks 1 & 2, the PGM distance $d(1,2) = d(2,1)$, but does this imply equal performance no matter what task is used as the base task? I.e., does it make a difference transferring from $1 \rightarrow 2$ and $2 \rightarrow 1$?
- Can this be explained by other means through rough linear correlation, say, based on the relevant fingerprints from the cheminformatics world?
- Is it possible that PGM may not be necessary at all, in case of perfect explainability by the above-mentioned linear model?

Response 2: Thanks for these constructive comments.

a) We have attempted to discuss some trends in Figure 2 and 3 based on the current literature and our understanding, as shown in Results in Line 254-287. We offer physical explanations where feasible and recognize that some patterns merit further investigation.

b) The symmetric presentation of Figure 2 reflects relative degrees of transferability rather than symmetric impact between molecule datasets, as I mentioned in Line 198-202: “When assessing the transferability from potential source datasets to a given target dataset, the distance between source and target datasets serves as an indicator of their knowledge transfer ability.” It is meaningful primarily when comparing potential sources for the same target dataset, or potential targets for the same source dataset. To clarify, when considering a target dataset A and source datasets B and C, if the PGM distance $d(B,A) > d(C,A)$, it indicates that source C is more related to target A. In other words, the transferability $C \rightarrow A$ is higher than $B \rightarrow A$. As demonstrated in Fig.3, for each target dataset, we compare the transferability from various source datasets to the target.

c) We appreciate the opportunity to clarify the distinction between the fingerprint-based linear correlation and PGM. As fingerprints in cheminformatics serve as a means of representing chemical compounds, they do not inherently contain information about the task. In contrast, PGM leverages gradient-based information that encompasses both the compound representation and the specific task. This provides a deeper insight into task relationships beyond what fingerprints and linear models can achieve. Therefore, PGM can’t be explained by rough linear correlation based on fingerprints.

d) While a linear correlation with fingerprints might offer a degree of explainability, it lacks the task-specific insights that are integral to understanding task-relatedness, as described in c). Hence, PGM is necessary.

Comment 3: Regarding the transfer model per se, it’s most commonly seen to be applied from one task to another, how about extending it to more tasks, or make a composite model from task 1 & 2, and transfer to task 3?

Response 3: Thank you for your insightful suggestion regarding the extension of transfer learning from one-to-one task transfer to encompass multiple tasks and the potential for creating composite models. We recognize the importance and potential benefits of such an approach in enhancing the generalizability and performance of models on downstream tasks. We have included this in our discussion of future work in Line 422-426: “(1) Extending the one-to-one transfer manner to simultaneously selecting the top-n similar source molecular property prediction datasets during the pre-training phase.”

Comment 4: One technical concern is about the explosion of the inverse of gradient (see eq. 4): how likely it may happen? And if it happens, how to fix it?

Response 4: Thanks for pointing out this issue. There is indeed a rare but possible risk of the gradient’s inverse exploding. To mitigate this and enhance the robustness of our approach, we employ the principal gradient calculated through expectation, rather than relying on the original gradient. Specifically, the proposed principal gradient is obtained through model re-initialization and gradient expectation calculation, defined by eq.1 as follows:

$$PGM(g_0) = E_{g_0}[\nabla \mathcal{L}(g_0)], \quad (1)$$

Afterwards, our method measures the transferability as the distance between the principal gradient obtained from model training on the source dataset and that derived from the target dataset. This process is mathematically formulated from eq.4 to eq.7 in Methods on Page 10.

REVIEWERS' COMMENTS:

Reviewer #1 (Remarks to the Author):

The authors have done an excellent job in addressing the comments of this reviewer. This reviewer has no additional comments.

Reviewer #2 (Remarks to the Author):

I believe all reviewer comments have been addressed thoroughly and this work should be published in its current form.

Reviewer #3 (Remarks to the Author):

The authors have resolved all my concerns, publish!

Response To Referees

Dear Editor and Reviewers,

Thank you for your constructive feedback and positive evaluation of our manuscript titled “Fast and effective molecular property prediction with transferability map” (Manuscript ID: COMMSCHEM-23-0558A). We are grateful for the opportunity to address the comments provided and are pleased that our revisions have met the approval of all three reviewers.

TO REVIEWER 1

Comment: The authors have done an excellent job in addressing the comments of this reviewer. This reviewer has no additional comments.

Response: We are pleased that the revisions met your expectations and are grateful for your supportive feedback.

TO REVIEWER 2

Comment: I believe all reviewer comments have been addressed thoroughly and this work should be published in its current form.

Response: We are encouraged by your endorsement and thankful for your detailed review.

TO REVIEWER 3

Comment: The authors have resolved all my concerns, publish!

Response: We are gratified by your approval and appreciate your constructive input throughout this process.

We have taken care to ensure our manuscript complies with the journal’s policies and formatting style, aiming to enhance the accessibility and impact of our work. We are looking forward to seeing our manuscript published in Communications Chemistry. Thank you once again for your time, feedback, and consideration.

Sincerely,

Corresponding author